**Data Availability Statement:** We consider our minimal underlying data set to contain sensitive data, and also potentially identifiable individuals

# Possible risk factors for poor asthma control assessed in a cross-sectional population-based study from Telemark, Norway

**Regine Abrahamsen[1], Gølin Finckenhagen Gundersen[1], Martin Veel Svendsen[1], Geir Klepaker[1,2], Johny Kongerud[2,3], Anne Kristin Møller Fell [1] ***

**1** Department of Occupational and Environmental Medicine, Telemark Hospital, Skien, Norway, **2** Institute of Clinical Medicine, Faculty of Medicine, University of Oslo, Oslo, Norway, **3** Department of Respiratory Medicine, Oslo University Hospital, Oslo, Norway

* annfel@sthf.no

## Abstract

This cross-sectional study of the general population of Telemark County, Norway, aimed to identify risk factors associated with poor asthma control as defined by the Asthma Control Test (ACT), and to determine the proportions of patients with poorly controlled asthma who had undergone spirometry, used asthma medication, or been examined by a pulmonary physician. In 2014–2015, the study recruited 326 subjects aged 16–50 years who had self-reported physician-diagnosed asthma and presence of respiratory symptoms during the previous 12 months. The clinical outcome measures were body mass index (BMI), forced vital capacity (FVC) and forced expiratory volume in one second ($FEV_1$), fractional exhaled nitric oxide (FeNO), immunoglobulin E (IgE) in serum and serum C-reactive protein (CRP). An ACT score $\leq$ 19 was defined as poorly controlled asthma. Overall, 113 subjects (35%) reported poor asthma control. The odds ratios (ORs) and 95% confidence intervals (CIs) for factors associated with poorly controlled asthma were: self-reported occupational exposure to vapor, gas, dust, or fumes during the previous 12 months (OR 2.0; 95% CI 1.1–3.6), body mass index $\geq$ 30 kg/m$^2$ (OR 2.2; 95% CI 1.2–4.1), female sex (OR 2.6; 95% CI 1.5–4.7), current smoking (OR 2.8; 95% CI 1.5–5.3), and past smoking (OR 2.3; 95% CI 1.3–4.0). Poor asthma control was also associated with reduced $FEV_1$ after bronchodilation (β –3.6; 95% CI –7.0 to –0.2). Moreover, 13% of the participants with poor asthma control reported no use of asthma medication, 51% had not been assessed by a pulmonary physician, and 20% had never undergone spirometry. Because these data are cross-sectional, further studies assessing possible risk factors in general and objectively measured occupational exposure in particular are needed. However, our results suggest that there is room for improvement with regards to use of spirometry and pulmonary physician referrals when a patient's asthma is inadequately controlled.

due to small groups. Sharing restrictions on the minimal data set are imposed from: The Regional Committee for Medical and Health Professional Research Ethics in South-east Norway (Study ID: REC 2012/1665), The Norwegian Data Inspectorate and the Telemark Hospital Department of Research and Development. However, data may be shared for researchers who meet the criteria for access to confidential data upon request to the head of the Telemark-Study steering committee: Dr. Trude K. Fossum, Department of Occupational and Environmental Medicine, Telemark Hospital, Post box 2900 Kjøbekk, 3710 Skien. E-mail: fotr@sthf. no The minimal data set identification (file name): 2020-02-05 Minimal data set PLOS ONE.sav

**Funding:** The funder of this study was the Department of Research and Development at Telemark Hospital, Skien, Norway; grant number 18.72, grant receiver Ms. Gølin Finckenhagen Gundersen. The funders had no role in study design, data collection, analysis, decision to publish, or preparation of the manuscript.

**Competing interests:** The authors have declared that no competing interests exist.

## Introduction

Asthma is the most prevalent chronic respiratory disease globally and imposes a substantial burden on patients, families, and communities [1, 2]. In particular, patients with severe asthma are hospitalized more often than other asthma patients, experience frequent exacerbations, and incur the majority of health care costs associated with this group of patients [3]. The Global Initiative for Asthma (GINA) guidelines state that asthma severity is a retrospective label that is assessed based on the treatment needed to control asthma, which in turn is assessed from two domains: symptom control and risk factors [2, 4].

Asthma control has been evaluated in a number of international studies, including several regions of Europe, in which both physicians and patients have reported poor levels of symptom control [5, 6]. These studies show that the prevalence of poor or suboptimal asthma control ranges from 57% to greater than 80%. This discrepancy is most likely related to the different methodologies applied, including the study group selected, because both selected patient populations [5] and samples from the general population [6] have been used. In the International Cross-Sectional and Longitudinal assessment of Asthma Control (LIAISON) study, major determinants for poor asthma control were reported to be seasonal worsening and persistent occupational exposure to allergens/irritants (self-reported and reported by their physician), followed by treatment-related issues. That study also reported that female sex, obesity, and smoking were associated with suboptimal asthma control. The Recognise Asthma and Link to Symptoms and Experience (REALISE) survey found that levels of asthma control were poor in a real-life sample from the general population of 11 European countries: 45% of respondents had uncontrolled asthma, and the level of well-controlled asthma ranged from 15% in Germany to 28% in Austria [6]. While these studies stressed the association between treatment-related issues and poor asthma control, little attention has been given to other important risk factors including occupational exposure to allergens and irritants [2, 4]. This information is considered to be important for improving work participation and asthma-related quality of life.

Our study aimed to assess the associations between possible risk factors and poor asthma control evaluated by the Asthma Control Test (ACT) [7, 8] in a sample of symptomatic asthma patients derived from a general population-based study in southeastern Norway. The following factors were assessed: self-reported exposure to occupational vapor, gas, dust, or fumes (VGDF), body mass index (BMI), sex, smoking, immunoglobulin E (IgE) in serum, serum C-reactive protein (CRP), fractional exhaled nitric oxide (FeNO), and lung function as assessed by spirometry. We also estimated the proportion of symptomatic asthma patients who had undergone spirometry, used asthma medication, or been examined by a pulmonary physician.

## Materials and methods

### Study sample

In February 2013, a random sample of 50,000 individuals aged 16–50 living in Telemark, a county in southeastern Norway, received a postal questionnaire as part of the Telemark study, which has been described in detail previously [9]. The response rate of the Telemark study was 33% (n = 16,099). Seven hundred non-responders were contacted by phone and/or mail and asked 13 key questions from the original questionnaire. Similar prevalence of physician-diagnosed asthma and several respiratory symptoms in responders and these non-responders were detected, although use of asthma medication was somewhat higher among those who responded (7.5% vs. 3.9% in non-responders).

As part of a nested case-control study conducted between August 2014 and December 2015, 1,857 asthma patients were eligible for medical examinations. Of these, 651 (35%) completed the medical examinations, 9% declined to participate, 34% did not attend their appointment, 11% had moved since 2013, and 10% lived more than 2 h by car from the medical examination locations and were therefore not invited. Altogether, 326 of those attending the medical examination for whom complete data were available reported having asthma symptoms during the previous 12 months. A flowchart of the participant selection process is shown in Fig 1.

## Questionnaire

The study participants were asked if they had ever had their lung function measured by spirometry ("Have you ever been examined by spirometry?"), whether they were using asthma medication ("Do you currently use medication for asthma?"), and whether they had ever been examined by a pulmonary physician ("Have you ever visited a pulmonary physician?"). Physician-diagnosed asthma, occupational VDGF exposure, and allergy was defined by an affirmative response to the following questions: "Has a physician ever diagnosed you with asthma?", and "Do you suffer from any form of allergy?".

Participants who gave a positive response to "Have you experienced an asthma attack during the past 12 months?", "Have you been awakened by heavy breathing/dyspnea at any time during the past 12 months?", or "Have you experienced whistling or wheezing in your chest at any time during the past 12 months?" were asked to complete the ACT [8]. The ACT contains questions about asthma symptoms and the use of asthma medication within the previous 4 weeks. In this sample derived from the general population, we chose not to include the group of subjects who reported physician-diagnosed asthma without any symptoms during the previous year. All participants were asked "Have you visited a doctor or accident/emergency unit because of acute breathing difficulties at any time in the past 12 months?", "Have you been hospitalized because of breathing difficulties at any time during the past 12 months?", and "Have you used extra cortisone medication or increased your cortisone inhalation at any time during the past 12 months?".

The single item question regarding self-reported occupational exposure to vapor, gas, dust or fumes (VGDF) was used: "Have you in your work been exposed to: vapor, gas, dust, or

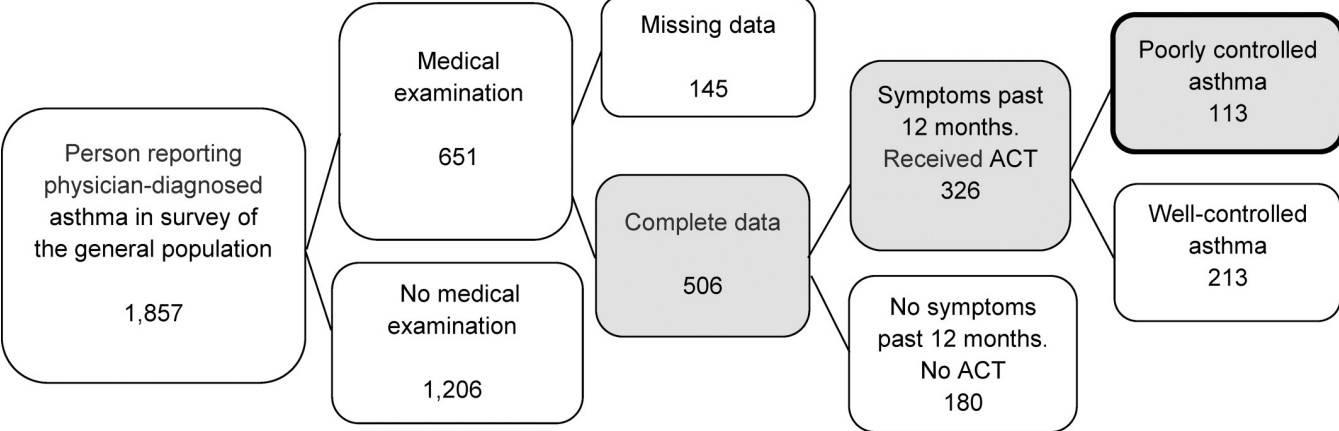

**Fig 1. Inclusion of participants reporting symptoms during the past 12 months (n = 326), and those with poorly controlled asthma (n = 113). ACT; Asthma control Test.**

fumes during the past 12 months?". The question shows good agreement with a multiple-item battery assessing such exposures, and modest agreement with a job-exposure matrix-based exposure categorization [10, 11].

## Obesity

Study team members measured all participants' body height and weight. Obesity was defined as BMI $\geq$ 30 kg/m$^2$ [12].

## ACT

The ACT is a widely used self-administered assessment tool to determine how well a patient's asthma is controlled [4, 7, 8]. The questionnaire consists of five questions regarding the occurrence of respiratory symptoms, medication use, and a self-assessment of symptom control during the previous 4 weeks. Each question is rated on a scale from 1–5 and values are summed for the ACT score. A low ACT score [5–19] indicates poorly controlled asthma, and a score of 20–25 indicates well-controlled asthma.

## Clinical variables

Lung function was assessed as part of the study in the period from August 2014–December 2015, using pre- and post-bronchodilator spirometry performed in accordance with the American Thoracic Society (ATS)/European Respiratory Society (ERS) guidelines with a Jaeger Master Screen PFT (Erich Jaeger GmbH & Co. KG, Würzburg, Germany) [13]. The spirometer was calibrated daily with a 3 L syringe. Forced vital capacity (FVC) as percent of predicted, forced expiratory volume in one second (FEV$_1$) as percent of predicted, and the FEV$_1$/FVC ratio were recorded. All tests were performed under the guidance of one of three trained physicians, and were manually validated by two trained physicians (GK and JK) according to ATS/ERS guidelines using flow–volume and time–volume curves [13]. For the analyses of the lung function indices, we selected those participants with at least one valid spirometry result. All reference values were calculated using the equations from the Global Lung Function Initiative guidelines [14]. The fraction of exhaled nitric oxide (FeNO) in exhaled air was included as a marker of eosinophilic inflammation, and measured according to the ATS/ERS criteria using a NIOX MINO (Aerocrine AB, Solna, Sweden) [15]. This device provides FeNO measurements at a 50 mL/s exhalation flow rate using an electrochemical sensor, with values expressed in parts per billion (ppb).

Peripheral blood was collected from all participants at the same visit as the performance of spirometry and FeNO and analyzed using standard procedures. The concentration of IgE was analyzed to assess allergic response using a Siemens Immulite 2000 XPI at the Department of Laboratory Medicine, Telemark Hospital, Skien. High-sensitivity CRP was included as a marker of systemic inflammatory response and analyzed using a Modul c702 Cobas 8000 modular analyzer (Roche Diagnostics) at the Department of Medical Biochemistry, Oslo University Hospital (Ullevål), Oslo.

## Statistical analyses

Pearson's chi-squared and Fisher's exact probability tests were used to compare categorical data, $t$-tests were used for normally distributed continuous data, and Mann–Whitney tests for non-normal continuous data. Multiple logistic regression was used to identify possible risk factors associated with poorly controlled asthma. Both crude and adjusted odds ratios (ORs) for other potential risk factors and confounders were calculated using a forward stepwise

regression, which resulted in a model including only the significant risk factors and confounders. Associations between asthma control and clinical variables (lung function, IgE, CRP, FeNO) were analyzed using linear regression analysis, adjusting for age, sex, education, smoking habit, and obesity. Due to skewed distribution of IgE, CRP and FeNO, the regression analysis were performed on the log-transformed variant of these variables. Collinearity was investigated by Pearson correlation showing weak correlations between the included proxies for socioeconomic status (education, smoking, obesity and VGDF). The strongest correlation was between education and smoking (r = 0.16). Sensitivity analyses were performed for lung function, blood samples, and FeNO without adjustment obesity, but did not alter the results. Further, stratification by age was performed but the groups were small and the confidence intervals overlapping (results not shown). Statistical analyses were performed using IBM SPSS Statistics (version 25; IBM SPSS, Armonk, NY, USA), and $p < 0.05$ was considered significant.

### Ethics approval

The Regional Committee for Medical and Health Professional Research Ethics (REC 2012/ 1665) approved the study. Participation was voluntary, and all participants were informed that they could withdraw from the study at any time without giving a reason. All participants signed an informed consent form. The study is registered at www.clinicaltrials.gov (NCT02073708).

### Results

The characteristics of subjects with physician-diagnosed asthma who had experienced respiratory symptoms during the previous 12 months (n = 326), stratified by well-controlled (n = 213) and poorly controlled asthma (n = 113), are presented in Table 1.

In the age group, 16–30; 36% of participants reported exposure to VGDF, while the percentage for those aged 31–40 and 41–50 were 27% and 24%, respectively.

Risk factors for poorly controlled asthma are presented in Table 2.

Women were more likely than men to have poorly controlled asthma (OR 2.6; 95% CI 1.5–4.7). Self-reported exposure to VGDF was associated with poor asthma control (OR 2.0; 95% CI 1.1–3.6), as was obesity (BMI $\geq$ 30 kg/m$^2$) (OR 2.2; 95% CI 1.2–4.1). Both past and current smoking were associated with poor asthma control (OR 2.3; 95% CI 1.3–4.0 and OR 2.8; 95% CI 1.5–5.3, respectively). The prevalence of having undergone spirometry, been examined by a pulmonary physician, and medication use, and the distribution of respiratory symptoms stratified by well-controlled and poorly controlled asthma, are shown in Table 3.

More frequent use of asthma medication and more healthcare visits because of recent breathing difficulties were seen among patients with poorly controlled asthma than among those with well-controlled asthma. Moreover, 20% and 51% of patients with poor asthma control had never undergone spirometry or been examined by a pulmonary physician, respectively. Twenty-four percent of participants with asthma symptoms during the previous 12 months, and 13% of those with poorly controlled asthma had not used asthma medication.

Linear regression analysis was performed to identify differences in clinical variables between poor and well controlled asthma cases (Table 4).

Table 4 shows that subjects with poor asthma control had post-bronchodilator FEV$_1$, while FeNO, IgE, and CRP were not statistically significant reduced.

### Discussion

In this sample from the general population, 35% of patients with asthma and respiratory symptoms during the previous 12 months reported having poorly controlled asthma, defined as an ACT score $\leq$ 19. Poor asthma control was associated with obesity, female sex, smoking and

**Table 1. Population characteristics.**

| | Received ACT (n = 326) | Well-controlled asthma (n = 213) | Poorly controlled asthma (n = 113) | Well-controlled vs. poorly controlled |
|---|---|---|---|---|
| | N (%) | N (%) | N (%) | P |
| Residential area | | | | |
| Urban | 220 (67) | 149 (70) | 71 (63) | 0.215* |
| Rural | 106 (33) | 64 (30) | 42 (37) | |
| Sex | | | | |
| Male | 104 (32) | 81 (38) | 23 (20) | **0.001***  |
| Female | 222 (68) | 132 (62) | 90 (80) | |
| Age (years) | | | | |
| 16–30 | 62 (19) | 40 (19) | 22 (19) | |
| 31–40 | 87 (27) | 64 (30) | 23 (20) | 0.361‡ |
| 41–50 | 177 (54) | 109 (51) | 68 (60) | |
| Education (years) | | | | |
| Elementary school (1–2) | 44 (13) | 24 (11) | 20 (18) | |
| Upper secondary and certificate (2–4) | 133 (41) | 83 (39) | 50 (44) | **0.026‡** |
| University ($\geq$ 4) | 149 (46) | 106 (50) | 43 (38) | |
| Smoking habits | | | | |
| Never smoker | 177 (54) | 133 (62) | 44 (39) | |
| Past smoker | 92 (28) | 51 (24) | 41 (36) | **<0.001†** |
| Current smoker | 57 (17) | 29 (14) | 28 (25) | |
| Body mass index (kg/m$^2$) | | | | |
| Normal weight ($\leq$ 24.9) | 119 (37) | 87 (41) | 32 (28) | |
| Overweight (25–29.9) | 111 (34) | 73 (34) | 38 (34) | **0.006‡** |
| Obese ($\geq$ 30) | 96 (29) | 53 (25) | 43 (38) | |
| Allergy | | | | |
| No | 92 (28) | 63 (30) | 29 (26) | 0.518* |
| Yes | 234 (72) | 150 (70) | 84 (74) | |
| Occupational VGDF previous 12 months | | | | |
| No | 245 (75) | 163 (77) | 82 (73) | 0.501* |
| Yes | 81 (25) | 50 (23) | 31 (27) | |
| | N (Poorly-/well-controlled)# | Median (IQR) | Median (IQR) | |
| Blood samples | | | | |
| IgE (Ku/L, ref** >87) | 323 (112/211) | 48 (128) | 74 (196) | 0.179§ |
| CRP (mg/L, ref** <5) | 323 (112/211) | 1.3 (2.0) | 2.0 (2.6) | **0.005§** |
| FeNO (ppb, ref** <25) | 307 (103/204) | 14.5 (14) | 11.0 (12) | **0.002§** |
| | N (Poorly/well controlled)# | Mean (SD) | Mean (SD) | |
| Spirometry | | | | |
| Pre-bronchodilator | | | | |
| FVC, % predicted | 311 (106/205) | 97.6 (11.7) | 94.0 (14.2) | **0.027¶** |
| FEV$_1$, % predicted | 311 (106/205) | 91.2 (14.4) | 86.9 (17.3) | **0.031¶** |
| FEV$_1$/FVC ratio in % | 311 (106/205) | 75.9 (8.0) | 75.0 (8.8) | 0.381¶ |
| Post-bronchodilator | | | | |
| FVC, % predicted | 279 (91/188) | 98.8 (11.1) | 97.0 (13.3) | 0.252¶ |
| FEV$_1$, % predicted | 279 (91/188) | 95.2 (13.2) | 92.0 (15.4) | 0.092¶ |
| FEV$_1$/FVC ratio in % | 279 (91/188) | 78.4 (7.6) | 77.2 (8.3) | 0.259¶ |

Statistically significant findings ($p < 0.05$) are in bold

* Fisher's exact probability test

**Reference values

† Pearson's chi-squared test

‡ Trend

§ Mann–Whitney test

¶ t-test

# Number of missing: IgE and CRP n = 3, FeNO n = 19, FVC, FEV$_1$, FEV$_1$/FVC ratio Pre-bronchodilator n = 15, Post-bronchodilator n = 47

ACT = asthma control test; VGDF = vapor, gas, dust, or fumes; FeNO = fraction of exhaled nitric oxide; IQR = interquartile range; SD = standard deviation

**Table 2. Logistic regression-estimated odds ratios for risk factors associated with poorly controlled asthma (n = 113).**

|  | OR$_{crude}$ (95% CI) | OR$_{adj}$ (95% CI)* | OR$_{adj}$ (95% CI)** |
|---|---|---|---|
| Residential area |  |  |  |
| Urban | 1.0 | 1.0 | NS |
| Rural | 1.4 (0.85–2.2) | 1.4 (0.81–2.3) |  |
| Sex |  |  |  |
| Male | 1.0 | 1.0 | 1.0 |
| Female | **2.4 (1.4–4.1)** | **2.6 (1.4–4.8)** | **2.6 (1.5–4.7)** |
| Age (years) |  |  |  |
| 16–30 | 1.0 | 1.0 |  |
| 31–40 | 0.65 (0.32–1.3) | 0.53 (0.25–1.2) | NS |
| 41–50 | 1.1 (0.62–2.1) | 0.83 (0.42–1.6) |  |
| Education (years) |  |  |  |
| Elementary school (1–2) | 1.0 | 1.0 |  |
| Upper secondary and certificate (2–4) | 0.72 (0.36–1.4) | 0.76 (0.35–1.6) | NS |
| University ($\geq$ 4) | **0.49 (0.24–0.97)** | 0.58 (0.26–1.3) |  |
| Smoking habits |  |  |  |
| Never smoker | **1.0** | **1.0** | 1.0 |
| Past smoker | **2.4 (1.4–4.1)** | **2.2 (1.2–3.9)** | **2.3 (1.3–4.0)** |
| Current smoker | **2.9 (1.6–5.4)** | **2.6 (1.4–5.2)** | **2.8 (1.5–5.3)** |
| Body mass index (kg/m$^2$) |  |  |  |
| Normal weight (18.5–24.9) | 1.0 | 1.0 | 1.0 |
| Overweight (25–29.9) | 1.4 (0.81–2.5) | 1.5 (0.83–2.9) | 1.6 (0.88–2.9) |
| Obese ($\geq$ 30) | **2.2 (1.2–3.9)** | **2.2 (1.2–4.1)** | **2.2 (1.2–4.1)** |
| Allergy |  |  |  |
| No | 1.0 | 1.0 | NS |
| Yes | 1.2 (0.73–2.0) | 1.3 (0.73–2.3) |  |
| Occupational VGDF previous 12 months |  |  |  |
| No | 1.0 | 1.0 | 1.0 |
| Yes | 1.2 (0.73–2.1) | **1.8 (1.0–3.4)** | **2.0 (1.1–3.6)** |

Statistically significant findings ($p < 0.05$) are in bold.

* Adjusted for all other variables in the model

** Adjusted only for significant variables using forward conditional regression. OR; odds ratio; CI; confidence interval; NS = not significant; VGDF = vapor, gas, dust, or fumes.

self-reported occupational VGDF exposure. Low asthma control was also associated with a small reduction in post-bronchodilatory FEV$_1$.. More than half (51%) of those reporting poor asthma control had not been examined by a pulmonary physician, 13% had not used asthma medication, and 20% had never undergone spirometry.

In this study, self-reported occupational VGDF was associated with poor asthma control (Table 2). Unfortunately, as in most population-based studies, objective measurements for occupational exposure were not available. However, the applied single item question regarding self-reported occupational exposure to vapor, gas, dust or fumes (VGDF), is commonly used in occupational epidemiology and has been tested against responses to a 16-item battery assessing specific inhalation exposures and against a job exposure matrix (JEM) [9, 10]. The authors concluded that the single VGDF survey item appears to delineate exposure risk at least as well as a multiple-item battery assessing such exposures [9], and shows modest agreement

**Table 3. Prevalence of spirometry, pulmonary physician examination, and medication use among those with physician-diagnosed asthma and symptoms during the previous 12 months.**

| | Received ACT* (n = 326) | Well-controlled asthma (n = 213) | Poorly controlled asthma (n = 113) | p-value** |
|---|---|---|---|---|
| Have you ever been examined by spirometry? | 253 (78%) | 163 (77%) | 90 (80%) | 0.578 |
| Have you ever visited a pulmonary physician? | 154 (47%) | 99 (47%) | 55 (49%) | 0.728 |
| Do you use asthma medication? | 246 (76%) | 148 (70%) | 98 (87%) | **0.001** |
| Have you experienced an asthma attack during the past 12 months? | 122 (37%) | 61 (29%) | 61 (54%) | **<0.001** |
| Have you been awakened by heavy breathing/dyspnea any time during the past 12 months? | 92 (28%) | 38 (18%) | 54 (48%) | **<0.001** |
| Have you experienced whistling or wheezing in your chest at any time during the past 12 months? | 229 (70%) | 134 (63%) | 95 (84%) | **<0.001** |
| Have you visited a doctor or accident/emergency unit because of acute breathing difficulties at any time during the past 12 months? | 52 (16%) | 18 (9%) | 34 (30%) | **<0.001** |
| Have you used extra cortisone medication or increased your cortisone inhalation at any time during the past 12 months? | 129 (40%) | 61 (29%) | 68 (60%) | **<0.001** |
| Have you been hospitalized because of breathing difficulties at any time during the past 12 months? | 4 (1%) | 2 (<1%) | 2 (2%) | 0.612 |

*ACT; Asthma Control Test.

** Fisher's exact test

with a JEM-based exposure categorization [9, 10]. Few studies have assessed occupation as a possible risk factor for poor asthma control [4], but our finding is consistent with those of previous studies reporting exacerbation of asthma from such exposure, and the LIAISON study, which found that self-reported occupational exposure to allergens/irritants was associated with poor asthma control [6, 16–18]. This cumulative evidence emphasizes the need for further efforts to reduce this possible risk factor, and for physicians to address occupational exposure in all asthma patients.

Obesity was also significantly associated with poor asthma control (Table 2). This observation is consistent with findings from several previous studies, and highlights the difficulty of

**Table 4. Linear regression to identify clinical differences between poor and well controlled asthma cases.**

| | N* (poor/well-control) | β (95% CI) | p-value |
|---|---|---|---|
| Pre-bronchodilator | | | |
| FVC-% predicted | 311 (106/205) | −3.7 (−6.7, −0.7) | **0.015*** |
| FEV$_1$% predicted | 311 (106/205) | −4.3 (−7.9, −0.6) | **0.022*** |
| FEV$_1$/FVC ratio in percent | 311 (106/205) | −0.8 (−2.7, 1.1) | 0.411* |
| Post-bronchodilator | | | |
| FVC % predicted | 279 (91/188) | −2.5 (−5.5, 0.5) | 0.105* |
| FEV$_1$% predicted | 279 (91/188) | −3.6 (−7.0, −0.2) | **0.036*** |
| FEV$_1$/FVC ratio in percent | 279 (91/188) | −1.0 (−2.8, 0.8) | 0.278* |
| Ln(IgE) | 323 (112/211) | 0.22 (−0.15, 0.59) | 0.251* |
| Ln(CRP) | 323 (112/211) | 0.09 (−0.13, 0.30) | 0.428* |
| Ln(FeNO) | 307 (103/204) | -0.08 (−0.25, 0.09) | 0.357* |

Statistically significant findings (p < 0.05) are in bold.

*Number of missing values: Pre-bronchodilator FVC, FEV1, FEV1/FVC ratio n = 15, Post-bronchodilator FVC, FEV1, FEV1/FVC ratio n = 47, IgE n = 3, FeNO n = 19.

**Adjusted for age, sex, education, smoking habits, obesity

achieving good asthma control in this group [6, 19–21]. Because weight loss may improve asthma control and lung function and reduce the need for medication in this group, our findings encourage the assessment of lifestyle factors in patients with poorly controlled asthma.

In this study, the OR for poor asthma control among women was more than twice that among men. This is in line with findings from a Swedish study from 2013 that showed that younger women had well-controlled asthma less often than men of the same age (OR 1.5; 95% CI 1.00–2.13) after adjusting for smoking, educational level, and BMI [22]. In a study from Saudi Arabia, 59% of men and 77% of women had uncontrolled asthma ($p = 0.002$) [23]. The international LIAISON study also reported that poor asthma control was associated with sex (men vs. women; OR 0.73; 95% CI 0.65–0.81) [6]. These findings highlight that special attention should be paid to women with poor asthma control.

Our results regarding smoking habits show that both past and current smokers may be more than twice as likely to have poorly controlled asthma than never smokers (Table 2). This is consistent with the GINA evidence showing that smoking exacerbates asthma, even in those with few symptoms [2]. Our findings may, in line with other studies presenting evidence of reduced asthma control and a greater need for health care among current smokers than among non-smokers and past smokers, emphasizes the importance of smoking cessation for asthma control. [24]

Table 1 shows that poor asthma control was associated with both an elevated level of the systemic inflammatory marker CRP and a reduced level of FeNO. After adjusting for possible confounders, CRP and FeNO were no longer significant (Table 4). This may imply that the univariate association is due to the confounders. Alternatively, the inclusion of not only severe asthma patients with signs of systemic inflammation or allergic response, but the whole range of subjects who had reported asthma symptoms in the previous 12 months could explain this finding.

Our results showed that post-bronchodilator $FEV_1$ was reduced in patients with poor asthma control (Table 4). A previous Swedish study reported that $FEV_1$ was associated with mortality [25], and a similar study from the US indicated that low $FEV_1$ was associated with increased mortality among patients with asthma [26]. However, in the Swedish study, post-bronchodilator tests were available only for parts of the cohort, while it is not clear whether the latter study used pre- or post-bronchodilator spirometry. In our study, both pre- and post-bronchodilator $FEV_1$ was associated with poor asthma control, whereas post-bronchodilator FVC was no longer significant after adjustment for possible confounders, including obesity. These findings combined underline the need for particular attention and close follow-up of patients with asthma who have reduced $FEV_1$.

To our knowledge, few studies have investigated the association between serum IgE and asthma control. In this study, no significant differences in IgE levels were observed between those with well- and poorly controlled asthma. Although a study from the US using data from the Severe Asthma Research Program found an inverse relationship between IgE levels and exacerbation [27], other studies have reported a positive association between IgE levels and asthma control [28, 29]. A possible explanation for these divergent results may be that some studies have evaluated the association between asthma control and a single measurement of total IgE, not longitudinal changes in asthma control and total IgE. Information regarding current treatment may also be important when interpreting total IgE levels in these patients [29]. Unfortunately, detailed information regarding treatment was not available in our study.

It is well known that patients with severe asthma may develop chronic airflow limitation [2]. The Norwegian guidelines for treatment of obstructive respiratory disease state that routine use of spirometry in general practice must be considered for patients at high risk of developing chronic airway obstruction [30]. Since the implementation of these guidelines, spirometry use in general practice increased from 24% in 1995/96 to 41% in 2003/04 [31]. A

2010 study from northern Norway showed that 70% of general practitioners used spirometry, consistent with our findings that 78% of our subjects had undergone spirometry (Table 2) [32]. International findings show large discrepancies in the use of spirometry by primary care services, from 6.7% in Australia to 42% in Belgium in 2011 [33]. Although spirometry use in Norway appears to be higher than that in many other countries, our results indicate that more than one in five patients with poorly controlled asthma may not have undergone spirometry.

Few studies have evaluated the proportion of patients with poorly controlled asthma who have been examined by a pulmonary physician. In our sample, only 49% of those with poorly controlled asthma had ever been assessed by a pulmonary physician. There are few specialist allergologists or severe asthma centers in Norway, hence, pulmonary physicians handle most cases with poor asthma control and severe asthma. According to the GINA guidelines, a lack of symptom control indicates the need for referral to a pulmonary physician or a severe asthma center to achieve better control and prevent disease progression [2]. We also observed a high prevalence of visits to a physician or accident/emergency department because of acute breathing difficulties among patients with poorly controlled asthma, supporting the need for specialist assessment. We found no significant difference between patients with well or poorly controlled asthma in terms of hospitalization because of respiratory problems. However, there were only four hospitalizations reported, so a larger sample size would be necessary to determine whether referral to a specialist leads to fewer hospitalizations and improved asthma control, as has been reported by others [2].

Twenty-four percent of participants who had asthma symptoms during the previous 12 months and 13% of those with poorly controlled asthma had not used any asthma medication. This is somewhat surprising because according to the Norwegian prescription register, Telemark County has been among the top five counties for use of asthma and COPD medication for several years, with an increase between 2012 and 2016 from 91 to 94 instances per 1000 inhabitants of all ages [34]. Potential reasons for the low use of asthma medication by the participants in this study may include that asthma patients are not using their prescribed medication, or that not all asthma patients with poor symptom control are assessed regularly or by a pulmonary physician; studies assessing asthma control over time are needed to clarify this issue.

There are a number of potential causes of poor symptom control in asthma, but our results suggest that patients with poorly controlled asthma may benefit from risk-factor evaluation and specialist assessment [35].

## Limitations

An important limitation in this study was that physician diagnosis of asthma was self-reported and could not be directly verified. However, the sensitivity and specificity of self-reported physician-diagnosed asthma has been validated [36], and it is widely used and regarded as well suited for epidemiological studies. To reduce the probability of chronic obstructive pulmonary disease (COPD), we restricted the study group to ≤50 years. However, self-report of asthma medication use, respiratory symptoms, and occupational VGDF exposure may have resulted in recall bias, which may have led to differential misclassification. Nevertheless, this study used the validated ACT combined with standardized and validated questionnaires about respiratory symptoms and diseases, which likely reduced the probability of misclassification [8, 36–38]. An important limitation of our study was that objective measurements of occupational exposure were not available. Hence, the observed association between occupational exposure and poor asthma control should be interpreted with caution.

The response rate of the Telemark-study, from which the present study population was derived, was relatively low (33%). Although assessment showed a slightly higher prevalence of

chronic cough and use of asthma medication among the study participants compared with non-responders, the prevalences of other respiratory symptoms and physician-diagnosed asthma were similar in participants and non-responders, indicating the validity of the estimates [11]. We also observed a relatively low response rate (35%) among those invited to the medical examinations. We have shown in a separate study that attendance to medical examinations was associated with BMI, sex, education, and smoking habits [39]. To decrease the likelihood of biased results, all regression analyses were adjusted for these factors. Further, it can be reasonably assumed that prevalence estimates would be more affected by increasing non-participation than associations between a risk factor and an outcome [40, 41]. Nevertheless, it is important to acknowledge that our sample size was limited, and that the results may not be entirely representative of an unselected population of people with symptomatic asthma.

We restricted the inclusion of participants to the age group 16 to 50 years. The youngest participants (16 to 30 years) reported the highest prevalence of occupational VGDF exposure (36% versus 27% and 24%). Although a substantial proportion of these subjects were unexposed students, apprentices and young workers may have a higher probability of exposure [42].

It could be argued that the analyses should have included all subjects with physician-diagnosed asthma, not only those who reported asthma symptoms within the previous 12 months. However, only 29% of the long-term asymptomatic subjects reported that they had used asthma medication. We considered it likely that asymptomatic subjects derived from a general population sample largely comprised those reporting childhood asthma without asthma symptoms in adulthood, and that the exclusion of these has probably resulted in somewhat lower estimates than if we had included all those who have ever been diagnosed with asthma.

Although our data are cross-sectional, they are from a sample of the general population and include all categories of patients with recent asthma symptoms, not only those with severe disease or a history of hospitalization. Importantly, as for all cross-sectional studies, no causal inferences could be drawn. This study was also limited in terms of size and geographic area; hence, larger, longitudinal studies are needed to confirm our findings.

## Recommendations

When assessing asthma control, we recommend an increased focus on patients who have occupational exposure, a history of tobacco smoking, are obese, or are women. In agreement with the Nordic consensus statement on the systematic assessment and management of possible severe asthma in adults, referral to a pulmonary physician or a severe asthma center should be considered when a patient's asthma is not well-controlled [35]. Consistent with the GINA strategy, we encourage physicians to use spirometry routinely to monitor all patients with asthma, particularly those with poorly controlled disease [2, 30].

## Conclusion

One in three patients with physician-diagnosed asthma reported poor asthma control as assessed by the ACT. Poorly controlled asthma was associated with self-reported occupational VGDF exposure, obesity, female sex, smoking, and reduced $FEV_1$. Given the uncertainty about the temporal sequence of events that is inherent in its cross-sectional design, and the lack of objective measurements of occupational exposure, this study could not evaluate any causal relationships between risk factors and asthma control. Nevertheless, our results may indicate that even in a country with a high level of social security and health services, there is room for improvement in the use of spirometry, asthma medication, and referral to a pulmonary physician, as well as assessment of possible risk factors for patients with poorly controlled asthma.

### Patient and public involvement

To realize the full potential of the study, we have involved user representatives in study planning, design piloting, and transfer of knowledge. A representative from The Norwegian Asthma and Allergy Association (NAAF) is a member of the study steering committee and has made valuable contributions to the development of questionnaires and methods for examination.

## Supporting information

**S1 File. Study questionnaire Norwegian.**
(PDF)

**S2 File. Study questionnaire English.**
(PDF)

## Acknowledgments

We thank the Blood Cell Research Group, Department of Medical Biochemistry, Oslo University Hospital and Department of Laboratory Medicine, Telemark Hospital, for the analysis of CRP.

## Author Contributions

**Conceptualization:** Regine Abrahamsen, Gølin Finckenhagen Gundersen, Martin Veel Svendsen, Johny Kongerud, Anne Kristin Møller Fell.

**Data curation:** Martin Veel Svendsen.

**Formal analysis:** Martin Veel Svendsen.

**Funding acquisition:** Anne Kristin Møller Fell.

**Investigation:** Regine Abrahamsen, Gølin Finckenhagen Gundersen, Martin Veel Svendsen, Geir Klepaker, Anne Kristin Møller Fell.

**Methodology:** Regine Abrahamsen, Gølin Finckenhagen Gundersen, Martin Veel Svendsen.

**Project administration:** Anne Kristin Møller Fell.

**Supervision:** Johny Kongerud, Anne Kristin Møller Fell.

**Validation:** Regine Abrahamsen, Gølin Finckenhagen Gundersen, Martin Veel Svendsen, Geir Klepaker, Johny Kongerud, Anne Kristin Møller Fell.

**Writing – original draft:** Regine Abrahamsen, Gølin Finckenhagen Gundersen, Johny Kongerud.

**Writing – review & editing:** Regine Abrahamsen, Gølin Finckenhagen Gundersen, Martin Veel Svendsen, Geir Klepaker, Johny Kongerud, Anne Kristin Møller Fell.

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
