## [Decision Letter · Decision Letter 0]

3 Feb 2020

PONE-D-19-34283

Poor asthma control associated with occupational exposure, body mass index, sex, and smoking in a cross-sectional population-based study from Telemark, Norway

PLOS ONE

Dear Dr Fell,

Thank you for submitting your manuscript to PLOS ONE. After careful consideration, we feel that it has merit but does not fully meet PLOS ONE’s publication criteria as it currently stands. Therefore, we invite you to submit a revised version of the manuscript that addresses the points raised during the review process.

We would appreciate receiving your revised manuscript by Mar 19 2020 11:59PM. To enhance the reproducibility of your results, we recommend that if applicable you deposit your laboratory protocols in protocols.io, where a protocol can be assigned its own identifier (DOI) such that it can be cited independently in the future. For instructions see: http://journals.plos.org/plosone/s/submission-guidelines#loc-laboratory-protocols

We look forward to receiving your revised manuscript.

Kind regards,

Davor Plavec

Academic Editor

PLOS ONE

Additional Editor Comments (if provided):

Please try to revise your manuscript according to the reviewers suggestions or write a rebuttal where you have arguments for it.

Journal Requirements:

Please provide an amended Funding Statement that declares *all* the funding or sources of support received during this specific study (whether external or internal to your organization) as detailed online in our guide for authors at http://journals.plos.org/plosone/s/submit-now.  Please state what role the funders took in the study.  If any authors received a salary from any of your funders, please state which authors and which funder. If the funders had no role, please state: "The funders had no role in study design, data collection and analysis, decision to publish, or preparation of the manuscript."

Reviewers' comments:

Reviewer's Responses to Questions

**Comments to the Author**

1. Is the manuscript technically sound, and do the data support the conclusions?

Reviewer #1: Partly

2. Has the statistical analysis been performed appropriately and rigorously? 

Reviewer #1: No

3. Have the authors made all data underlying the findings in their manuscript fully available?

Reviewer #1: No

4. Is the manuscript presented in an intelligible fashion and written in standard English?

Reviewer #1: Yes

5. Review Comments to the Author

Reviewer #1: This was a very interesting study, which was well-written. Here are my specific observations and suggestions:

1. Line 38: how likely is it that the youngest participants were exposed to occupational fumes? It would be helpful to provide an additional table stratifying the descriptive statistics by age category due to heterogeneity in the sample.

2. Sensitivity analysis conducted by age group. Stratify the analysis by age groups due to the different stages of lung development across the age range. It is possible that some older participants have COPD, for example, and thus poorer lung function. Furthermore, teenagers are likely to under/mis-report smoking habits.

3. Education, smoking, obesity and possibly VGDF are likely to be proxies for socio-economic status and all correlated. The authors should investigate multicollinearity in their models and the relationships between these variables before including them in the regression models.

4. Line 164: “adjustment obesity” - please add “for”.

5. Table 1: Poorly controlled asthma for age: %s do not add to 100.

6. What are the reference/normal ranges for the clinical variables? This would provide greater clarity about differences and severity between both groups.

7. Greater clinical context is required about what the clinical variables mean and why they are relevant. To reader with limited clinical knowledge, it isn’t clear what the measures imply given the results. For example, there is no discussion about what CRP is, why it has been included in the study, and what the results mean clinically.

8. Table 4 and Methods: This is very confusing and unclear. I thought that poor asthma control (a binary variable) is the outcome throughout the paper (as also implied by the title of Table 4). If this is the case, your model is incorrect. Linear regression is for continuous outcomes. In light of these comments, the analysis and discussion surrounding Table 4 is unclear. Clarity is required on why you have chosen the model, what is the hypothesised causal relationship, and which variables are the dependent/independent variables.

9. The median and IQR has been reported for some clinical variables, suggesting they are not normally distributed. The authors should check whether the assumptions of the linear regression model have been upheld and whether these variables require transformation (e.g., logged).

10. Line 249: The authors should investigate whether non-significance of clinical variables could be due to the distribution of these variables.

11. Results have been presented for the first time in the Discussion section. I suggest that these should be presented in the Results section.

6. PLOS authors have the option to publish the peer review history of their article (what does this mean?). If published, this will include your full peer review and any attached files.

Reviewer #1: No

---

## [Author Response · Author response to Decision Letter 0]

2 Mar 2020

Point to point response to editors and reviewers’ comments:

First, we would like to thank the reviewer for the thorough review and valuable comments that have allowed us to adjust accordingly, and improve the manuscript significantly. 

Journal Requirements:

We have now labelled the files according to the style template.

We have now included both a Norwegian and an English copy of the questionnaire as Supporting Information. 

a. Please provide an amended Funding Statement that declares *all* the funding or sources of support received during this specific study (whether external or internal to your organization) as detailed online in our guide for authors at http://journals.plos.org/plosone/s/submit-now. 

This is now included in the cover letter.

b. Please state what role the funders took in the study. If any authors received a salary from any of your funders, please state which authors and which funder. If the funders had no role, please state: "The funders had no role in study design, data collection and analysis, decision to publish, or preparation of the manuscript." Please include your amended statements within your cover letter; we will change the online submission form on your behalf.

This is now included in the cover letter. 

4. In your Data Availability statement, you have not specified where the minimal data set underlying the results described in your manuscript can be found. PLOS defines a study's minimal data set as the underlying data used to reach the conclusions drawn in the manuscript and any additional data required to replicate the reported study findings in their entirety. All PLOS journals require that the minimal data set be made fully available. For more information about our data policy, please see http://journals.plos.org/plosone/s/data-availability. Upon re-submitting your revised manuscript, please upload your study’s minimal underlying data set as either Supporting Information files or to a stable, public repository and include the relevant URLs, DOIs, or accession numbers within your revised cover letter. For a list of acceptable repositories, please see http://journals.plos.org/plosone/s/data-availability#loc-recommended-repositories. Any potentially identifying patient information must be fully anonymized. Important: If there are ethical or legal restrictions to sharing your data publicly, please explain these restrictions in detail. Please see our guidelines for more information on what we consider unacceptable restrictions to publicly sharing data: http://journals.plos.org/plosone/s/data-availability#loc-unacceptable-data-access-restrictions. Note that it is not acceptable for the authors to be the sole named individuals responsible for ensuring data access. We will update your Data Availability statement to reflect the information you provide in your cover letter.

This is also included in the cover letter.

5. Review Comments to the Author

Reviewer #1: This was a very interesting study, which was well-written. Here are my specific observations and suggestions:

1. Line 38: how likely is it that the youngest participants were exposed to occupational fumes? It would be helpful to provide an additional table stratifying the descriptive statistics by age category due to heterogeneity in the sample.

We have performed the stratification on age for the results in table 1. We have chosen not to include these in a new table because the table would be considerably larger and the important and interesting results are the following: “In the age group, 16-30; 36% of participants reported exposure to VGDF, while the percentage for those aged 31-40 and 41-50 were 27% and 24%, respectively.” This sentence is now included in the results section.

2. Sensitivity analysis conducted by age group. Stratify the analysis by age groups due to the different stages of lung development across the age range. It is possible that some older participants have COPD, for example, and thus poorer lung function. Furthermore, teenagers are likely to under/mis-report smoking habits.

As suggested, we have performed sensitivity analyses (stratification) by age. However, the groups were small and the confidence intervals overlapping (results not shown) and thus to our opinion no meaningful information could be derived from these analyses. However, we have included the following to the statistic section: “ Further, stratification by age was performed but the groups were small and the confidence intervals overlapping (results not shown).”

In addition, we added the following regarding COPD and young participants (regarding the latter a new reference was added nr. 42) to the limitation section of the discussion: 

1. “To reduce the probability of chronic obstructive pulmonary disease (COPD), we restricted the study group to ≤50 years.”

2. “We restricted the inclusion of participants to the age group 16 to 50 years. The youngest age group (16 to 30 years) reported the highest prevalence of occupational VGDF exposure (36%). Although a substantial proportion of these subjects were unexposed students, apprentices and young workers may have a higher probability of exposure [42].” 

3. Education, smoking, obesity and possibly VGDF are likely to be proxies for socio-economic status and all correlated. The authors should investigate multicollinearity in their models and the relationships between these variables before including them in the regression models.

The following has been included in the statistics section: “Collinearity was investigated by Pearson correlation showing weak correlations between the included proxies for socioeconomic status (education, smoking, obesity and VGDF). The strongest correlation was between education and smoking (r=0.157).” 

Because income is not necessarily a good measure of socioeconomic status in Scandinavia, education is often used. We have choose also to adjust for smoking because of the association with asthma severity. We have shown in other studies (based on the same cohort) that asthma is associated with obesity and exposure to VGDF. Hence, the results were adjusted for these four variables. 

4. Line 164: “adjustment obesity” - please add “for”.

Thank you, this has been corrected.

5. Table 1: Poorly controlled asthma for age: %s do not add to 100.

This is due to rounding of numbers.

6. What are the reference/normal ranges for the clinical variables? This would provide greater clarity about differences and severity between both groups.

The reference values/range are now included in table 2. 

7. Greater clinical context is required about what the clinical variables mean and why they are relevant. To reader with limited clinical knowledge, it isn’t clear what the measures imply given the results. For example, there is no discussion about what CRP is, why it has been included in the study, and what the results mean clinically.

We have met this comment by inclusion of the following information in the methods section (the new text is underlined) : Measurement of the concentration of IgE was included to assess allergic response and analyzed using a Siemens Immulite 2000 XPI at the Department of Laboratory Medicine, Telemark Hospital, Skien. High-sensitivity CRP was included as a marker of systemic inflammation and analyzed using a Modul c702 Cobas 8000 modular analyzer (Roche Diagnostics) at the Department of Medical Biochemistry, Oslo University Hospital (Ullevål), Oslo. 

Accordingly, the following adjustment was made regarding FeNO: The fraction of exhaled nitric oxide (FeNO) in exhaled air was included as a marker of eosinophilic inflammation, and was measured according to the ATS/ERS criteria using a NIOX MINO (Aerocrine AB, Solna, Sweden) [13]. 

In addition, the following adjustment (the new text is underlined) is made to the corresponding text in the discussion section: Table 1 shows that poor asthma control was associated with both an elevated level of the systemic inflammatory marker CRP and a reduced level of FeNO. After adjusting for possible confounders, CRP and FeNO were no longer significant (Table 4). This may imply that the univariate association is due to the confounders. Alternatively, the inclusion of not only severe asthma patients with signs of systemic inflammation or allergic response, but the whole range of subjects who had reported asthma symptoms in the previous 12 months could explain this finding. 

8. Table 4 and Methods: This is very confusing and unclear. I thought that poor asthma control (a binary variable) is the outcome throughout the paper (as also implied by the title of Table 4). If this is the case, your model is incorrect. Linear regression is for continuous outcomes. In light of these comments, the analysis and discussion surrounding Table 4 is unclear. Clarity is required on why you have chosen the model, what is the hypothesised causal relationship, and which variables are the dependent/independent variables.

Thank you for this important comment. We agree that the table title and text was unclear. Linear regression was used to identify differences in clinical variables between poor and well controlled asthma cases. The clinical variables were the dependent in these analyses. We have now changed the title of table 4 and the text referring to this table in the results and discussion sections accordingly. 

9. The median and IQR has been reported for some clinical variables, suggesting they are not normally distributed. The authors should check whether the assumptions of the linear regression model have been upheld and whether these variables require transformation (e.g., logged).

We have now performed the analyses on log transformed variables for IgE, CRP and FeNO (table 4). The conclusions are upheld, but we agree that the log transformed variables should be used and have included them in table 4. 

10. Line 249: The authors should investigate whether non-significance of clinical variables could be due to the distribution of these variables.

Log transformed variables for IgE, CRP and FeNO are now used and did slightly influence the numbers, but the results are still not statistical significant (table 4). Please see also our response to comment 9.

11. Results have been presented for the first time in the Discussion section. I suggest that these should be presented in the Results section.

Thank you, we have now also included the following to the text in the results section: “Twenty-four percent of participants with asthma symptoms during the previous 12 months, and 13% of those with poorly controlled asthma had not used asthma medication.” 

Sincerely yours

Anne Kristin M. Fell, MD, PhD

Corresponding author

---

## [Decision Letter · Decision Letter 1]

3 Apr 2020

PONE-D-19-34283R1

Poor asthma control associated with occupational exposure, body mass index, sex, and smoking in a cross-sectional population-based study from Telemark, Norway

PLOS ONE

Dear Dr Fell,

Thank you for submitting your manuscript to PLOS ONE. After careful consideration, we feel that it has merit but does not fully meet PLOS ONE’s publication criteria as it currently stands. Therefore, we invite you to submit a revised version of the manuscript that addresses the points raised during the review process.

Based on significant arguments of the reviewer the manuscript needs major revision. The single question used in your study to assess possible occupational exposure can not be the proper argument to use this in both the Title and throughout the article. If you remove this kind of statement there is not much left regarding your methodology. Please revise your manuscript according to the reviewers remarks but be careful about your statements. They have to be based on proper argumentation and scientific proof.

We would appreciate receiving your revised manuscript by May 18 2020 11:59PM. To enhance the reproducibility of your results, we recommend that if applicable you deposit your laboratory protocols in protocols.io, where a protocol can be assigned its own identifier (DOI) such that it can be cited independently in the future. For instructions see: http://journals.plos.org/plosone/s/submission-guidelines#loc-laboratory-protocols

We look forward to receiving your revised manuscript.

Kind regards,

Davor Plavec, MD, MSc, PhD, Prof.

Academic Editor

PLOS ONE

Additional Editor Comments (if provided):

Based on significant arguments of the reviewer the manuscript need major revision. The single question used in your study to assess possible occupational exposure can not be the proper argument to use this in both the Title and throughout the article. If you remove this kind of statement there is not much left regarding your methodology. Please revise your manuscript according to the reviewers remarks but be careful about your statements. They have to be based on proper argumentation and scientific proof.

Reviewers' comments:

Reviewer's Responses to Questions

**Comments to the Author**

1. If the authors have adequately addressed your comments raised in a previous round of review and you feel that this manuscript is now acceptable for publication, you may indicate that here to bypass the “Comments to the Author” section, enter your conflict of interest statement in the “Confidential to Editor” section, and submit your "Accept" recommendation.

Reviewer #2: All comments have been addressed

2. Is the manuscript technically sound, and do the data support the conclusions?

Reviewer #2: No

3. Has the statistical analysis been performed appropriately and rigorously? 

Reviewer #2: Yes

4. Have the authors made all data underlying the findings in their manuscript fully available?

Reviewer #2: Yes

5. Is the manuscript presented in an intelligible fashion and written in standard English?

Reviewer #2: Yes

6. Review Comments to the Author

Reviewer #2: Respecting the comments from the previous review, the authors competently answered all the questions asked and made the required changes in the course of the manuscript.

Unfortunately, while reading the manuscript, I noticed some significant weaknesses.

Specifically, the title, methods, results, discussion, and conclusions of the article should be changed because it refers to poor asthma control associated with occupational exposures, ... which is actually based on a conclusion drawn from the answer to just one question asked in the questionnaire " Have you been exposed to: vapor, gas, dust or fumes during the past 12 months? ".

Therefore, no objective measurements were carried out or occupational exposure was objectively evaluated based on a validated questionnaire, as well.

A major revision of the article is needed, and the same should be pointed out in limitation in the study.

It is also necessary to validate the questionnaire.

The final number of respondents for the "population study" is also doubtful.

7. PLOS authors have the option to publish the peer review history of their article (what does this mean?). If published, this will include your full peer review and any attached files.

Reviewer #2: No

---

## [Author Response · Author response to Decision Letter 1]

16 Apr 2020

Point to point response to editors and reviewers’ comments:

First, we would like to thank the editor for the possibility to revise the manuscript a second time, and the reviewer for the additional comments that have allowed us to adjust accordingly and improve the manuscript. 

Editors comment: 

Based on significant arguments of the reviewer the manuscript needs major revision. The single question used in your study to assess possible occupational exposure can not be the proper argument to use this in both the Title and throughout the article. If you remove this kind of statement there is not much left regarding your methodology. Please revise your manuscript according to the reviewers remarks but be careful about your statements. They have to be based on proper argumentation and scientific proof.

We agree that the best way to assess occupational exposure is by performing objective measurements of the agents involved. However, this is often not feasible in population-based studies. We have changed the title and adjusted the manuscript throughout to meet this important comment. We apologize for not including references regarding the validation of the single-item question regarding occupational exposure, this has now been included. The manuscript has been changed throughout regarding this issue, the most important changes are listed below (I-IV): 

I: To underline the limitations of a single item question regarding occupational exposure, we have added the following to the limitation part (p.18): “An important limitation of our study was that objective measurements of occupational exposure were not available. Hence, the observed association between occupational exposure and poor asthma control should be interpreted with caution.”

II: We have added the following to the methods section including references (p. 7): “The single item question regarding self-reported occupational exposure to vapor, gas, dust or fumes (VGDF) was applied: ““Have you in your work been exposed to: vapor, gas, dust, or fumes during the past 12 months?”. The question shows good agreement with a multiple-item battery assessing such exposures, and modest agreement with a job exposure matrix-based exposure categorization [9, 10].”

II: We have also added the following text to the discussion (p. 13-14): “Unfortunately, as in most population-based studies, objective measurements for occupational exposure were not available. However, the applied single item question regarding self-reported occupational exposure to vapor, gas, dust or fumes (VGDF), is commonly used in occupational epidemiology and has been tested against responses to a 16-item battery assessing specific inhalation exposures and against a job exposure matrix (JEM) [9, 10]. The authors concluded that the single VGDF survey item appears to delineate exposure risk at least as well as a multiple-item battery assessing such exposures [9], and modest agreement with a JEM-based exposure categorization [9 ,10].“ 

We would like to add that the applied VGDF question has been used also in more recent studies [1, 2].

1. Paulin LM, Smith BM, Koch A, et al. Occupational Exposures and Computed Tomographic Imaging Characteristics in the SPIROMICS Cohort. Ann Am Thorac Soc 2018;15(12):1411-1419.

2. Murphy D, Bellis K, Hutchinson D. Vapour, gas, dust and fume occupational exposures in male patients with rheumatoid arthritis resident in Cornwall (UK) and their association with rheumatoid factor and anti-cyclic protein antibodies: a retrospective clinical study. BMJ Open 2018;8:e021754. doi:10.1136/bmjopen-2018-021754.

IV: Finally, the conclusion has been changed (the new text is underlined): “Given the uncertainty about the temporal sequence of events that is inherent in its cross-sectional design, and the lack of objective measurements of occupational exposure, this study could not evaluate any causal relationships between risk factors and asthma control.”

Reviewer 2:

Respecting the comments from the previous review, the authors competently answered all the questions asked and made the required changes in the course of the manuscript.

Unfortunately, while reading the manuscript, I noticed some significant weaknesses.

1. Specifically, the title, methods, results, discussion, and conclusions of the article should be changed because it refers to poor asthma control associated with occupational exposures, ... which is actually based on a conclusion drawn from the answer to just one question asked in the questionnaire " Have you been exposed to: vapor, gas, dust or fumes during the past 12 months? ". Therefore, no objective measurements were carried out or occupational exposure was objectively evaluated based on a validated questionnaire, as well. A major revision of the article is needed, and the same should be pointed out in limitation in the study. 

Thank you for this important comment. We have changed the manuscript throughout accordingly (including the limitation section). Please see our reply to the editor above. 

2. It is also necessary to validate the questionnaire.

The applied questionnaire has been validated, this was addressed in our manuscript (limitations p. 17): “Nevertheless, this study used the validated ACT combined with standardized and validated questionnaires about respiratory symptoms and diseases, which likely reduced the probability of misclassification [8, 36–38].“ 

Regarding validation of the single item question addressing occupational VGDF exposure, we have added more information. Please see also our response to the editors’ comment above. 

3. The final number of respondents for the "population study" is also doubtful.

We agree with the reviewer that the final number of respondents is limited. We have addressed non-response in the limitation part of the discussion (p. 18): “The response rate of the Telemark-study, from which the present study population was derived, was relatively low (33%). Although assessment showed a slightly higher prevalence of chronic cough and use of asthma medication among the study participants compared with non-responders, the prevalences of other respiratory symptoms and physician-diagnosed asthma were similar in participants and non-responders, indicating the validity of the estimates [11]. We also observed a relatively low response rate (35%) among those invited to the medical examinations. We have shown in a separate study that attendance to medical examinations was associated with BMI, sex, education, and smoking habits [39]. To decrease the likelihood of biased results, all regression analyses were adjusted for these factors. Further, it can be reasonably assumed that prevalence estimates would be more affected by increasing non-participation than associations between a risk factor and an outcome [40, 41].”

On page 19 the problem of our limited sample size was addressed: “This study was also limited in terms of size and geographic area; hence, larger, longitudinal studies are needed to confirm our findings.” 

To meet the reviewers comment, and to further underline the limited sample size, we have now added the following (p. 18, new text is underlined): “Nevertheless, it is important to acknowledge that our sample size was limited, and that the results may not be entirely representative of an unselected population of people with symptomatic asthma.”

We hope that we have been able to address the comments in a satisfactory way and are looking forward to your response.

Sincerely yours

Anne Kristin M. Fell, MD, PhD

Corresponding author

---

## [Editor Report · Decision Letter 2]

20 Apr 2020

Possible risk factors for poor asthma control assessed in a cross-sectional population-based study from Telemark, Norway

PONE-D-19-34283R2

Dear Dr. Fell,

We are pleased to inform you that your manuscript has been judged scientifically suitable for publication and will be formally accepted for publication once it complies with all outstanding technical requirements.

With kind regards,

Davor Plavec, MD, MSc, PhD, Prof.

Academic Editor

PLOS ONE

Additional Editor Comments (optional):

After making a suggested revisions to your manuscript the manuscript is acceptable for publication in its current form. Thanks for working with our reviewers and our editors on improving the quality of your manuscript.
---

## [Editor Report · Acceptance letter]

23 Apr 2020

PONE-D-19-34283R2 

Possible risk factors for poor asthma control assessed in a cross-sectional population-based study from Telemark, Norway 

Dear Dr. Fell:

I am pleased to inform you that your manuscript has been deemed suitable for publication in PLOS ONE. Congratulations! Your manuscript is now with our production department. 

With kind regards,

on behalf of

Dr. Davor Plavec 

Academic Editor

PLOS ONE